# Decision-Making Regarding On-Farm Culling Methods for Dairy Cows Related to Cow Welfare, Sustainable Beef Production, and Farm Economics

**DOI:** 10.3390/ani15111651

**Published:** 2025-06-03

**Authors:** Mariska Barten, Yvette de Geus, Joop den Hartog, Len Lipman

**Affiliations:** 1KIWA CMR, 3436 ZZ Nieuwegein, The Netherlands; mariska.barten@kiwa.com; 2Department Population Health Sciences, Faculty of Veterinary Sciences, Institute for Risk Assessment Sciences, Universiteit Utrecht, 3584 CS Utrecht, The Netherlands; y.degeus@uu.nl; 3Ministry of Agriculture, Fisheries, Food Security and Nature, 2594 AC The Hague, The Netherlands; jt.denhartog5@gmail.com

**Keywords:** on-farm emergency slaughter (OFES), welfare, culling, euthanasia, meat salvage

## Abstract

Decision-making regarding the on-farm culling of dairy cows is complex. Dairy farmers can choose from various methods, such as prolonged veterinary treatment, regular transport to a slaughterhouse, on-farm emergency slaughter, or euthanasia on the farm. The results of this study demonstrate that different regulations and various factors can influence individual dairy farmers’ decision-making processes, including their preferences and eventual choice of culling methods. Considerations regarding culling methods often reveal conflicting interests, such as farm economics, animal welfare, and sustainable beef production. Understanding the factors that influence decision-making in this context enables stakeholders—such as governments, the dairy and meat industries, cattle traders, and private veterinary practitioners—to develop tools that address these competing interests and provide more targeted support to dairy farmers in their decision-making.

## 1. Introduction

The number of dairy cows in the Netherlands was recently estimated to be 1.5 million, with an average yearly replacement of 22.7% of all cows [1]. Most of these are slaughtered and account for around 42% of the total beef production in the Netherlands [2]. Dairy cows that are removed from the herd are often indicated as cull cows. The definition of “culling” is defined in different ways in the literature [3,4]. In this article, “culling” refers to the removal of cows from the herd for sale, (emergency) slaughter, or disposal due to mortality (including unassisted death and euthanasia). Culling is carried out for a broad variety of reasons, such as low production, health problems, and failure to conceive [5,6,7]. The culling mode chosen by a dairy farmer is partly dependent on the reason for culling. According to the European Transport Regulation, regarding the protection of animals during transport and related operations, injured, weak, and diseased animals may not be transported, as it is assumed that the transport will cause added suffering [8]. Cattle unable to stand, move painlessly, or walk unaided are unfit for transport. The size of the group of dairy cows not fit for transport is substantial, since lameness is an important reason for culling [9,10,11,12].

For a limited group, on-farm emergency slaughter (OFES) and mobile slaughterhouses (MSHs) can be considered. OFES is only applicable for healthy animals that have suffered an accident that prevented transport to the slaughterhouse for welfare reasons [13]. In the Netherlands, these animals are only eligible for OFES up to 72 h after the accident occurred [14]. Longer existing local abnormalities or abnormalities that are not caused by an accident, such as many types of lameness, do not meet the criterium of “accident” and, therefore, exclude the option for OFES.

Between 2015 and 2019, the yearly number of OFES was approximately 9000 cattle in the Netherlands [15], and in 2022 and 2023, around 12,000 cattle were subjected to OFES each year [16].

Mobile slaughterhouses (MSHs) could potentially be used for a wide range of dairy cows that are unfit for transport but are assessed fit for human consumption (BuRO) [17,18]. National competent authorities must approve the use of these facilities, which, in the Netherlands, is conducted according to set conditions [19]. In contrast to the application of MSHs in other countries, the target population in the Netherlands is cattle with health conditions. Because no processing of carcasses after stunning and exsanguination was carried out, the facility was referred to as a mobile slaughter unit (MSU) [20]. In the period between December 2018 and April 2020, an MSU was available for Dutch farmers as part of a pilot. Ante-mortem (AM) inspections were performed by official veterinarians of the NVWA on the farms. After the cows were stunned and exsanguinated inside the MSU (or in the barn if the cow was unable to walk to the MSU), these exsanguinated slaughtered cows were transported to a slaughterhouse. Although added welfare value for a substantial number of cows was recognized, no subsequent initiatives were started after the pilot in 2019. Under current EU legislation and national conditions, exploitation of MSU and MSHs is considered economically unviable [20,21,22]. In the Netherlands, around 52,000 dairy cows die on the primary farm each year due to natural death or euthanasia [23]. Part of this group of cows is assumed eligible for human consumption but is euthanized due to the lack of on-farm slaughter methods. It is estimated that the target group of cattle not fit for transport but assessed eligible for human consumption amounts to at least 20,000 animals per year in the Netherlands [17,18,19,20]. Euthanasia is indicated to prevent further suffering, especially when a cow is in pain and has a limited chance of recovery [24,25,26,27]. Dutch legislation [28] states that livestock farmers are required to provide the necessary care for their animals. Although other interests may conflict with the choice of culling method, animal welfare remains a key factor in the decision-making process.

This study focuses on conflicting interests and the decision-making process regarding culling of dairy cattle with health conditions resulting in exclusion from regular livestock transport to slaughterhouses. Semi-structured interviews were conducted with dairy farmers, private veterinary practitioners (PVPs), slaughterhouse operators, and livestock traders to examine the motivations behind different culling methods for cows. These stakeholders were specifically asked for their opinions on the extent to which on-farm killing methods prior to slaughter, such as OFES and MSUs, could contribute to or undermine animal welfare.

## 2. Materials and Methods

Semi-structured interviews were conducted by the first author in various regions of the Netherlands between October 2022 and April 2024. Different stakeholders who play a role in the decision-making process of culling cows were identified: farmers (herd size between 60 and 390 cows older than 2 years, average 185), veterinarians (specialized in dairy cattle), livestock traders, and slaughterhouse operators. In the Netherlands, livestock traders often consult with the slaughterhouse operator on options for transport and slaughter of cattle. The selection criteria for the participants mainly focused on geographical characteristics and business activities related to culling of dairy cows. The recruitment method, participant, and company numbers are described in Table 1.

Some PVPs were contacted using the personal network of other participants and of the first author. One PVP and one operator of a slaughterhouse that receives many cull cows in the Netherlands took the initiative to register for participation after they became aware of the study. Most dairy farmers were suggested by the participating PVPs from their client registers. Six slaughterhouses that proportionately receive many emergency slaughters in the Netherlands were specifically invited to participate by the first author. These slaughterhouses became known through information shared by veterinarians, livestock traders, and dairy farmers in the interviews. The first author contacted eligible participants by telephone to explain the purpose of the research. Following these calls, one dairy farmer declined participation, while all other contacted individuals agreed to take part in the study. During the initial contact, some livestock traders were excluded from participation as they did not appear to be the appropriate target group, indicating that they primarily traded in other animal species or groups rather than cull cows.

After agreement for participation, background information about the research project was provided by e-mail. Interview subjects and questions were sent to the participants in advance (list of questions in Appendix A). Interviewees gave their verbal consent to audio recording and anonymous processing of their opinions and information they shared. While answering the questions, the participants were not guided by options. Beyond the predetermined questions, the participants were invited to share their own input relevant to the subject. The length of the interviews varied between 60 and 120 min.

The number of participants was determined by theme saturation per professional group [29]. No selection was made based on the age or gender of the participants. Ten of the fifty participants were female (1 slaughterhouse operator, 7 veterinarians, and 2 dairy farmers).

The recorded files were listened to and transcribed in detail by the first author. The interview transcripts were uploaded into the NVivo 14 software program. After coding the information, common answers and themes emerged. Any information in the transcripts that related to a specific person or veterinary practice was omitted or replaced by non-identifiable descriptors. The interviews were analyzed using the grounded theory research method described by Chun Tie et al., 2019 [30]. The grounded theory is a well-known methodology employed in many research studies. In a grounded theory study, qualitative and quantitative data generation techniques can be used. Grounded theory sets out to discover or construct theory from data, systematically obtained and analyzed using comparative analysis. Using this method, patterns and recurring themes were identified in the interview data.

## 3. Results

### 3.1. Decision Making Process Regarding (Prolonged) Veterinary Treatment and Different Culling Modes, Such as (OFES) Slaughter and Euthanasia, for Dairy Cows

Almost all participants (dairy farmers, livestock trader, and PVPs) indicated that the decision-making process regarding (prolonged) veterinary treatment and different culling modes, such as (on-farm emergency) slaughter and euthanasia (Figure 1), is complex. Various factors such as estimated prognosis, economic and emotional value of an individual cow, fitness for transport, deemed eligibility for human consumption, costs of veterinary treatments, withdrawal times for milk and meat of veterinary medicines, quality of life assessment, possibilities, and willingness of the farmer to house and milk a dairy cow individually for a longer period come into consideration in the decision-making process. Additionally, participants from all groups indicated spontaneously that requirements regarding mortality rates imposed by dairy companies or private quality labels for raw milk delivery [31,32] also influence the decision-making process. Most dairy farmers try to avoid high mortality rates in this context because it can cause obligations from dairy companies to make plans to improve animal health. Suppliers to private quality labels for raw milk deliveries may lose their surplus on the basic price for raw milk when specific requirements, such as those regarding mortality rates, are not met. The requirements regarding mortality rates appear to be a more important financial incentive in the considerations than the expected meat yield.

In general, decisions regarding culling are not made from a purely economic perspective, as noted by dairy farmers and PVPs. Farmers mentioned, for example, the genetic and emotional value of the animal. Investment and time constraint were also seen as important. Many livestock traders and PVPs emphasized that differences between situations, farms, and farmers exist when it comes to weighing factors and uncertainties in the decision-making process.

### 3.2. Reasons to Prefer (On-Farm Emergency) Slaughter Above Euthanasia

All PVPs and livestock traders emphasized that dairy farmers generally prefer OFES or regular slaughter instead of euthanasia for dairy cows that have lost their primary economic value for milk production when they are assessed eligible for human consumption. This preference is usually driven, at least partly, by economic interests. Euthanasia and rendering cost money, while regular slaughter or OFES generally generates some money by meat yield. However, the meat yield of dairy cows is often negligible in relation to the turnover on dairy farms, particular for on-farm emergency slaughtered cows. Farmers explicitly mentioned that the potential loss of the financial surplus on raw milk prices or the obligation to draw up an improved health plan when mortality rates become too high is important in the decision-making process. By sending cows for slaughter, these risks can be avoided.

The participants also mentioned non-economic factors that contributed to the preference of many dairy farmers to cull cows for (on-farm emergency) slaughter. Most PVPs and dairy farmers stated that they feel an ethical aversion to euthanize cows deemed eligible for human consumption. Most participants stated that OFES honored the lives of these cows and contributed to sustainable beef production goals. In general, many dairy farmers reported being reluctant to make decisions about euthanasia. Watching an animal die or being euthanized is usually distressing for the owner and/or caretaker. In addition to the dairy farmers themselves, some PVPs and livestock traders also report that some farmers are sensitive to social judgment from their surrounding environment when a dead cow is on their premises. Some dairy farmers indicated that they perceive the on-farm death of a cow as a personal failure.

### 3.3. Reasons to Prefer Euthanasia Above (On-Farm Emergency) Slaughter

The interviewed PVPs indicated that euthanasia was most frequently requested for cows deemed not eligible for human consumption or not fit for livestock transport, or in situations where the requirements for OFES are not met. For farmers, this choice was also driven by economic considerations, since price penalties will likely to be imposed when cows are (unexpectedly) assessed not fit for transport or not eligible for OFES by the official veterinarian at the slaughterhouse. Costs for trading, transport, slaughter, ante-mortem (AM) and post-mortem (PM) inspection, and rendering will also arise when a cow is assessed not eligible for human consumption at PM inspection. In some cases, even when a cow’s fitness for transport and suitability for human consumption are unquestioned, farmers still prefer euthanasia for various non-economic reasons. Some dairy farmers chose euthanasia to avoid any potential involvement of the NVWA, even if the risk was minimal. Others felt reluctant to cull certain individual cows for slaughter, for example, when the cow had produced a high milk yield throughout her life. These farmers wanted to save the cow the stress of removal and transport to the slaughterhouse, even when it meant forgoing the expected meat yield, to prioritize animal welfare. Transport distances were also a consideration for some dairy farmers. When regional slaughter options were unavailable, euthanasia was sometimes preferred (for individual cows) over transporting them to a slaughterhouse at a greater distance. However, the interviews revealed that most farmers were unaware of the routes and transport distances travelled by cull cows. Livestock traders noted that only a small number of dairy farmers showed interest in these aspects and generally did not interfere with the traders’ decisions. Most dairy farmers did not make agreements with livestock traders regarding trading conditions, such as transport distances. A high personal quality standard regarding meat quality served as an argument for a small number of dairy farmers to opt for euthanasia. Euthanasia was executed when these farmers decided not to consume the meat themselves. Finally, the PVPs and livestock traders noted that a small group of dairy farmers more frequently chose euthanasia, since they never opted for OFES due to their aversion to the procedures of stunning and exsanguination.

### 3.4. Assigned Value of On-Farm Killing Methods Prior to Slaughter

The value of on-farm killing methods prior to slaughter, such as OFES and MSHs, was emphasized by nearly all participants across various professional groups. The reasons included non-economic factors on farms, such as avoidance of welfare issues and improving cow welfare by avoiding stress during transport. At slaughterhouses, the emphasized reasons were salvaging meat in line with sustainability goals and reducing risks related to the image of practices in dairy farms, livestock transport, and the dairy and beef industries.

The costs associated with OFES and MSU are often close to the expected revenue of the meat yield, indicating that dairy farmers have limited economic interest when dairy cows are eligible for human consumption but unfit for transport. According to slaughterhouse operators, from an economic standpoint, the timing of OFES should not be delayed if cows are deemed eligible for human consumption. Operators stressed that the risk of losing meat value or having the meat declared unfit for human consumption rises quickly over time. They explained that cows could lose body weight rapidly when they are in discomfort or pain. Furthermore, meat value decreases when cows are unable to stand for long periods, leading to significant muscle damage. Opportunities for OFES or MSH, therefore, encouraged effective decision-making. PVPs, slaughterhouse operators, and livestock traders suggested that decisions based on farm business economics often aligned with non-economic interests as well, such as preventing cow welfare issues and avoiding negative perceptions of farm operations. However, most participants were not positive about the viability of MSH under the current conditions.

### 3.5. Decision-Making Related to On-Farm Killing Methods

Stakeholders from all interviewed groups believed that skills in identifying animal health problems early, skills in veterinary treatments, and different culling modes all contribute to prevention of animal health and welfare problems. Reducing the number of lame dairy cows is regarded as an important area for improvement by both farmers and veterinarians. In addition, improvements in housing, such as the adjustment of cubicles and preventing slippery floors, were mentioned by veterinarians and livestock traders as measures to prevent accidents. For cows assumed not eligible for human consumption, it was considered important that costs for euthanasia and requirements from dairy companies and private dairy delivery programs regarding mortality rates did not inhibit timely euthanasia decisions. Almost all participants believed that mortality rate requirements should be adjusted in a way that does not hinder sustainable and animal-friendly decisions.

## 4. Discussion

### 4.1. Mortality and On-Farm Euthanasia of Cattle

Danish research into culling modes of 281,286 dairy cows demonstrated that 83.3% of these cows were slaughtered, 10.2% died unassisted, and 6.5% were euthanized [33]. In the Netherlands, around 52,000 dairy cows die on the primary farm each year unassisted or by euthanasia [23]. The incidence of mortality in dairy production systems has increased over the last decades [3,34] and is made worse by strict enforcement of transport legislation and the lack of options for on-farm killing methods prior to butchery.

Early detection of cow health issues, correct medical interventions, and proactive culling decisions can help prevent significant welfare problems [35,36,37,38] and reduce on-farm mortality. However, culling decisions are complex. Farmers have various goals and may prioritize these differently [39]. Herd health programs and PVPs can assist dairy farmers in adopting preventive measures to reduce lameness, for example, or other health conditions that make cull cows unfit for transport. They can also support farmers in the decision-making process. PVPs can help increase on-farm awareness, promote prevention efforts to reduce accidents, and provide management support for injury and (early) culling decisions [12,35,38,40,41,42].

On-farm killing methods prior to slaughter can help prevent welfare issues in cattle with health problems that are deemed eligible for human consumption but unfit for transport. However, these methods do not address concerns related to the end-of-life situations of cattle assessed as not fit for human consumption. In such cases, euthanasia may be necessary to prevent severe and irreversible suffering [43,44]. The degree of suffering is an important factor in decision-making regarding euthanasia, but assessing this degree is not considered easy. This aligns with the fact that cattle, as prey animals, do not typically show obvious signs of distress [45], which increases the risk that euthanasia may not be performed in a timely manner. The tendency of cattle to suffer in silence should be recognized by veterinarians and dairy farmers, as timely euthanasia is a critical factor in ensuring cow welfare [46]. Delays or mistakes in performing euthanasia can negatively impact cow welfare and also affect the emotional well-being of those carrying out the procedure [25,47]. The result of this study aligns with the results from other studies demonstrating that decision-making regarding euthanasia can be complex and difficult for dairy farmers [39,44,48,49].

Both the literature and our study show that decision-making becomes increasingly complex when a cow has a health condition that results in no clear signs of suffering but has no prospect of recovery within the economic model of the dairy farm. This presents a more difficult decision for a farmer and PVP than euthanizing a cow that is clearly severely ill. In such cases, euthanasia may be considered appropriate for economic reasons. This type of decision is likely the most challenging for the farmer and the PVP, and it is less well accepted by the public [44]. Veterinary costs for euthanasia also appear to influence the timing of the procedure. The way these costs are incurred should be carefully evaluated by all stakeholders. Establishing clear, fixed rates for veterinary euthanasia services would benefit both the welfare of the cows and the business interests of PVPs.

Protecting the reputation of the dairy and beef sectors is crucial for all stakeholders in the meat industry. Therefore, improved mortality registration on dairy farms—such as recording the cause of death with a clear distinction between unassisted death and euthanasia—would be valuable. It is important to recognize that cows dying unassisted are likely to suffer the most [35,39]. Detailed mortality records can also contribute to better herd health and help prevent misrepresentations of individual dairy farms and the dairy industry when mortality rates are used for benchmarking.

### 4.2. Different Perspectives Related to On-Farm Emergency Slaughter in Europe

Perspectives on OFES vary across European countries. For example, a study in Norway showed that more than 4% of all cattle slaughtered were dealt with using OFES, and 8% of the country’s dairy cows were subjected to this form of slaughter. Considerable market support mechanisms and the availability of 24/7 services for OFES likely contributed to its favorability [50]. Additionally, the differences can be attributed to Norway’s non-EU status, allowing for it to adapt EU legislation to a certain extent. For example, the term “accident” has been translated as “unforeseen event”, a phrase that likely permits a broader interpretation than the original European legislation allows [50,51]. The Norwegian guidelines stress that OFES should take place as soon as possible following an unforeseen event, with the only exception being specific lameness issues. In contrast to regulations in the Netherlands, dairy cows with a prolapse uteri, milk fever, or obstetrical problems are also eligible for OFES in Norway [51]. Finally, official veterinarians in Norway do not have to be present when cattle are killed, as is mandatory in the Netherlands and Finland [50]. Within the EU, differences in the practice of OFES exist. For instance, OFES in Malta dramatically reduced the number of cattle assessed as unfit for transport arriving at slaughterhouses [52]. Furthermore, in Italy, the EU regulations related to OFES are implemented differently from the Netherlands, resulting in the situation that most cattle deemed unfit for transport are eligible for OFES [53]. Interestingly, OFES is rare in France, with only 432 cases reported across the country in 2020. Farmers in France seem reluctant to have their animals slaughtered on-farm, with severely injured animals routinely being transported to slaughterhouses [54]. Instructions in France specify that a PVP should first assess a cow that has suffered an accident. The PVP then determines whether the cow can be transported, provided certain conditions are met, such as additional bedding and transport to the nearest slaughterhouse within 48 h of the accident. As in 2015, the European Commission in 2022 recommended that France should implement measures to ensure that only fit animals are transported to slaughterhouses.

The European Parliament supports on-farm killing methods prior to slaughter, such as OFES and MSHs, since these initiatives are valued as beneficial for animal welfare. Additionally, MSHs are regarded positively in the context of regional/short chain food production [21,55,56,57]. As in the Netherlands, the contribution of OFES to the prevention of welfare issues is also recognized by Irish dairy farmers [58]. However, costs for OFES and MSHs must not significantly exceed the expected meat yield for the application to remain feasible. The costs for stunning and exsanguination, (individual) transport of the slaughtered cow to the slaughterhouse, PM inspection, and potential sampling and testing for bovine spongiform encephalopathy (BSE) have risen in recent years. This has been partly due to the decline in the number of slaughterhouses in the Netherlands. The OFES service, like in Ireland, is not available in every region of the Netherlands and is only offered on specific days and times. Funding small abattoirs to provide the OFES service could enhance availability, thereby improving animal welfare [58]. Additionally, farmers’ motivations to transport severely injured animals to slaughterhouses could be reduced, and meat salvage could be improved.

The decline in the number of slaughterhouses in the Netherlands also impacts the welfare of regular cull cows transported for slaughter, as it leads to longer transport distances. In most countries, the target group of cattle for MSHs may also be fit for regular transport to a slaughterhouse. Therefore, while meat salvage can be seen as an additional benefit in the Netherlands, it may not necessarily be the case in other countries [20].

### 4.3. Requirements Regarding Mortality Rates Imposed by Dairy Companies or Private Quality Labels for Raw Milk

Transporting cows with health issues in order to minimize mortality on dairy farms should be avoided because it risks animal welfare, violates regulations, compromises food safety, and harms the reputation of the dairy and beef industries. Unintended side effects on animal welfare and food safety due to requirements regarding mortality rates imposed by dairy companies or private quality labels for raw milk should, therefore, be eliminated.

### 4.4. Meat Salvage in the Context of Sustainability Objectives

The reduction in food loss and waste has become increasingly important from economic, environmental, and social perspectives. In 2019, approximately 18 billion animal lives were embodied in losses and waste during global meat production and consumption [59]. Euthanasia of farm animals assessed eligible for human consumption is considered a form of food loss [60]. Skúladóttir et al. (2022b) estimated that up to 40% of the meat from cows that die on dairy farms each year could be salvaged for human consumption, highlighting the need for solutions to address meat waste [50]. Furthermore, euthanized animals have to be treated at a rendering plant, which can cause environmental constraints.

Although our study specifically focused on decision-making in dairy cattle, it is likely that similar welfare concerns and meat salvage considerations also apply to other farm animal species and other groups of cattle. To reduce food loss, optimal slaughter facilities are needed. MSH and OFES are seen as valuable initiatives in the prevention of situations that harm cow welfare and meat salvage [36]. These initiatives should, therefore, be viewed from a broader perspective and may need a mindset change in stakeholders. At the moment, the operating costs of OFES and MSH are generally too high to be attractive to dairy farmers. For young cattle and smaller species like pigs, sheep, and goats assessed eligible for human consumption but unfit for transport, these initiatives are even less attractive due to the lower expected meat yield. Meat yield can vary depending on meat quality and fluctuating meat prices per kilogram. To ensure the initiative can reach a wide range of livestock, both variable and fixed costs, such as those for transport, slaughter, and ante- and post-mortem inspections, should not significantly exceed the expected meat yield.

There are concerns regarding the economic viability and availability of MSH and OFES, as highlighted by Dutch stakeholders in this study and also acknowledged in other countries [36]. To support cow welfare, meat salvage, dairy farm economics, and the reputation of the dairy and meat industries, MSH and OFES options could be beneficial. In this context, a critical evaluation of the conditions for OFES could help to address unintended bottleneck effects caused by the wording of regulations at the EU and national levels. At the EU level, it is stated that cows must have suffered an accident preventing their transport to the slaughterhouse to meet the conditions for OFES. While the European Commission likely intended this condition to prevent cows unfit for human consumption from being delivered to slaughterhouses, adjustments in the wording would increase the number of cattle eligible for OFES and fit for human consumption. Furthermore, additional national regulations and rules should be critically assessed to ensure they align with the intent of EU law. Other options for cattle with health conditions but deemed fit for human consumption, such as (short) direct individual transport with precautions to a regional slaughterhouse, should also be considered.

### 4.5. Limitations of This Study

The methodology used in this study is based on the grounded theory method. The quality of a grounded theory is based on three principles: firstly, the researcher’s expertise, knowledge, and research skills; secondly, the methodological congruence with the research question; and, finally, procedural precision in the use of the method [30]. Data collection and analytical conceptualization need to be rigorous throughout the research process to secure excellence in the final grounded theory.

Several research limitations must be considered when interpreting the findings. Qualitative research seeks to understand the nature of phenomena and does not allow for quantifiable measures or the inferences from quantitative methods. Because of the design of this study, only a limited number of representatives could be interviewed per professional group. However, the participants’ views on the various topics were largely similar, regardless of which professional group the participant represented. The results may, therefore, be seen as largely representative. The opinions of livestock traders and slaughterhouse operators were seen as very valuable because of their overview of the meat production chain and the number of animals they encounter.

Markedly, more men than women participated in this study. Similar studies also saw a higher participation of men compared to women [58]. It is assumed that this fact had a minimal impact on our findings, as men are more prevalent than women in the relevant professional groups that participated.

Due to the given limitations in terms of the selection of participants and the sensitivity of the information shared, it is possible that the problems regarding cow welfare may be more serious than outlined here.

## 5. Conclusions

In the decision-making process related to culling cows not suitable for transport, many factors and interests are considered by farmers, veterinarians, livestock traders, and slaughterhouse operators. These interests—such as cow welfare, private quality assurance factors, food safety, and meat salvage—often conflict, which can result in animal welfare constraints. OFES and the introduction of a mobile slaughterhouse could help to solve these constraints. Furthermore, these conflicts can contribute to an increased risk of non-compliance with legislation at various stages in the dairy and meat production chain. For future studies, it would be important to examine whether expanding the conditions outlined in Dutch and EU legislation regarding OFES could lead to increased food safety risks.

## Figures and Tables

**Figure 1 animals-15-01651-f001:**
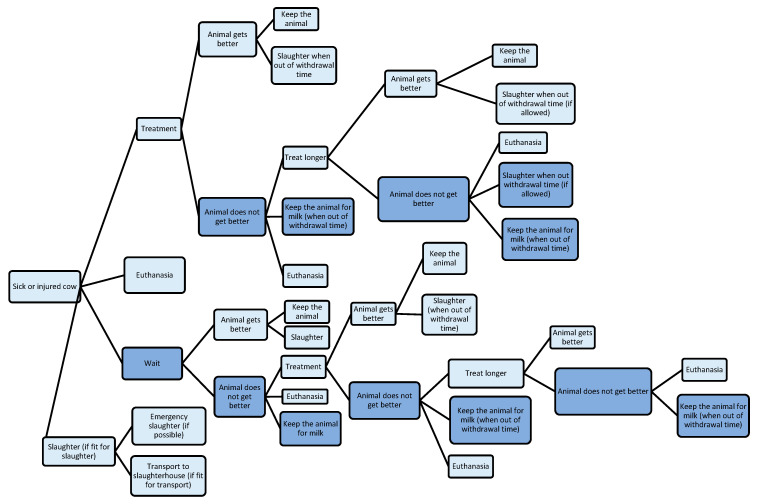
Decision tree with considerations for treatment with veterinary medicines and culling for (on-farm emergency) slaughter or euthanasia. Dark blue boxes: possible negative effects on animal welfare.

**Table 1 animals-15-01651-t001:** Recruitment and characteristics of the participants in the interviews.

	Number of Participants *n* (%)	Number of Organizations *n* (%)	Method of Recruitment
Dairy farmers	14 (25%)	14 (32%)	Proposed by private veterinary practitioners
Private veterinary practitioners	23 (42%)	16 (36%)	Internet research
Livestock traders	7 (13%)	7 (16%)	Internet research
OFES * slaughterhouse operators	11 (20%)	7 (16%)	Approached by first author
Total	55 (100%)	43 (100%)	

* OFES; on-farm emergency slaughter.

## Data Availability

The raw data supporting the conclusions of this article can be made available anonymously upon request from the authors.

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
