# Peer review of "Decision-Making Regarding On-Farm Culling Methods for Dairy Cows Related to Cow Welfare, Sustainable Beef Production, and Farm Economics"

_animals, 2025, doi:10.3390/ani15111651_

Round 1
Reviewer 1 Report
Comments and Suggestions for Authors
The end-of-life for dairy cattle is a very current and important topic. This paper addresses the real conflicts that arise in dairy farming at the time of decision-making on culling of dairy cows that are not suitable for transport according to the actual EU and national regulations. The issue of conflicts between regulations and the real needs of the sector, with potential negative effects on cattle welfare, is very relevant. Specific comments on the manuscript follow.
Abstract:
L17 and L23: Please delete the comma after 'which factors' and after 'OFES', respectively.
LL23-25: You can hypothesize this, but you showed no data that objectively demonstrated it. Actually, cow welfare is compromised when culling is necessary but is delayed. In your paper, you analyzed which factors play a role in decision-making about culling, but you did not clearly report the proportion of cows that suffered for "delayed" culling. Please revise.
Material and Methods:
I'm not very familiar with the Grounded theory approach, but I have some doubts about the composition of the participant sample:
1) I think you should better explain the competencies and importance of the opinions of the various categories of participants to the decision-making process of culling cows unsuitable for transport (e.g., I suppose that slaughterhouse operators have little to do with this process, despite they can tell us something about meat suitability for human consumption)
2) You have quite an equal number of slaughterhouse operators and dairy farmers, I think you should discuss a bit how their opinions were weighted in the process of data analysis.
3) The number of dairy farmers that participated in this study was quite low with respect to the other categories and to their principal role in the decision-making process. I'm aware that you cannot improve it, but can you at least describe the average herd size (with min and max) the interviewed farmers conducted?
4) Also for veterinaries, it would be useful to know how many farms they followed, in order to better understand their level of experience in the field.
5) I understand that you run semi-structured interviews, but can you provide the list of questions you used?
Results:
L149: Please replace the abbreviation 'eg' with 'like' or another synonym.
LL163-164: 'Suppliers to private quality labels for raw milk deliveries suppliers to private quality concepts may lose their surplus on the basic price for raw milk when specific requirements' - Maybe there is a writing mistake (repetition), please check.
LL168-171: Why didn't you report direct farmers' opinions here? More in general: is it possible to have a ranking of importance for factors influencing culling-mode decisions?
LL177-187: Again, why farmers' direct opinion is not mentioned?
L195: You reported quantitative information only here (i.e., '2/14' of farmers), can you provide more throughout the Results? This would help the reader understand the weight of the opinions.
L235: Brackets for 'limited' are unnecessary, please delete them.
L252: The word 'animal' here sounds like a repetition, please remove it.
Discussion:
I suggest totally inverting the order of the paragraphs from 4.1 to 4.4 (i.e., 4.4 = 1; 4.3 = 2; 4.2 = 3; 4.1 = 4), as the current last one (4.4) directly deals with the results of the study, and the first one is a more general discussion.
Regarding meat salvage (paragraph 4.1), what about 'second choice meat' and/or meat to be destined for pets? Sometimes meat of cows that need to be culled is not perfectly suitable for human consumption, but could be saved for pet carnivores, as their food is usually thermally treated or pasteurized. Is there any reference that considers this destination for meat?
L302: 'industry reputation' is repeated, please correct.
L362: Please shift the citation '(Royal GD, 2021)' at the end of the sentence.
L416: Please add a comma after 'research skills'.
Conclusions:
LL434-436: I don't think your study demonstrated what you stated here. You didn't collect any data that can support this statement. Actually, you determined which factors are considered by different actors in the decision-making process about culling of cows that are not suitable for transport. Then, you have identified some potential risks related to the conflicts that arise in the decision-making processes. Please revise.
Author Response
Dear Editor,
We would like to thank the reviewers for all the work they have put in. With their help the aricle has much improved.
Kind regards,
Len Lipman

Reviewer 2 Report
Comments and Suggestions for Authors
Please see review attached.

Author Response
See below

Reviewer 3 Report
Comments and Suggestions for Authors
Thank you very much for this article on such an important and timely topic. It highlights critical issues related to animal welfare and the ethical dilemmas faced by all stakeholders involved.
The manuscript presents highly valuable and relevant content, particularly in its in-depth exploration of stakeholder perspectives and regulatory complexity in dairy cow culling decisions. However, the readability of the text should be improved by adding tabular or graphical summaries that would help readers quickly grasp the key findings. Otherwise, some of the most insightful results risk being overlooked or underappreciated by the audience.
I would like to recommend to strengthen the paper by a tabular presentation of interview responses - relevance to content quality by summarizing the responses by stakeholder group (e.g., farmers, veterinarians, traders, slaughterhouse operators) significantly improves clarity and transparency. It allows for easy side-by-side comparison of perspectives — e.g., who prefers OFES and why and helps readers quickly grasp key themes without needing to parse long narrative paragraphs.
Especially in qualitative research using Grounded Theory, a structured table helps demonstrate systematic data analysis and supports the coding process more clearly.
A table summarizing cross-country regulations and practices related to on-farm killing—particularly on-farm emergency slaughter (OFES)—would not only be helpful, but also methodologically and substantively highly valuable.
Readers can quickly grasp the extent to which national regulations differ (e.g., how “accident” is defined, applicable time limits, veterinary oversight).
It also clearly illustrates how the Netherlands compares within the broader European context.
A table reinforces the argument that EU regulations are interpreted inconsistently across Member States, sometimes leading to contradictions or unintended effects.
Policy Relevance
For stakeholders and policymakers, such a comparative overview would be particularly useful in the context of regulatory assessment and potential reform proposals.
Comments on the Quality of English Language
While the manuscript provides a strong and valuable contribution, several stylistic and grammatical refinements were necessary throughout the text to improve clarity and flow to ensure that the quality of writing matches the strength of the content.
I have taken the liberty of making a few initial adjustments and would be very happy if these are considered helpful and appropriate. I kindly suggest continuing the revision of the remaining sections in a similar manner, should you find this approach suitable.
Line 13-31:
In the Netherlands, approximately 52,000 dairy cows die on farms each year due to natural causes, euthanasia, or on-farm emergency slaughter (OFES). Farmers are responsible for deciding on the most appropriate course of action, often in consultation with a veterinarian, livestock trader, or slaughterhouse operator. To identify the factors influencing this decision-making process, semi-structured interviews were conducted with dairy farmers, private veterinary practitioners, livestock traders, and slaughterhouse operators. Dairy cattle culling decisions are shaped and constrained by strict livestock transport regulations and limited availability of on-farm killing methods. Additionally, mortality rate requirements imposed by the dairy industry and private quality labels for raw milk deliveries affect farmers' decisions. Furthermore, restrictive regulations concerning OFES and mobile slaughterhouses (MSHs) can have unintended negative impacts on cow welfare and meat salvage. The interests of stakeholders—such as cow welfare, food safety, economic concerns, industry reputation, and sustainability goals—frequently conflict. This study demonstrates that the decision-making process around culling or (prolonged) veterinary treatment is complex, involving a careful weighing of various factors, interests, and uncertainties that can differ between farms and farmers
Line 36: In 2023, the number of dairy cows …
Line 40-41: is variably defined in the literature
Line 41-43: “culling” refers to the removal of cows from the herd for sale, (emergency) slaughter, or disposal due to mortality (including unassisted death and euthanasia)
Author Response
Dear editor,
I have included in this reaction the reply to reviewer 3.
Kind regards,
Len Lipman

Round 2
Reviewer 1 Report
Comments and Suggestions for Authors
I appreciated the efforts made by the Authors to improve the manuscript.
I still have only minor comments:
Mat&Met:
Despite the Authors' declaration that a list of questions used in the questionnaire was now provided, I couldn't see it in the new draft of the paper. I suggest providing either a summary description of the questions in the Mat&Met section, or a more detailed list as Supplementary Material.
Table 1:
I suggest reporting the new details provided on farmers (i.e., herd size) and Veterinarians (i.e., dairy cattle specialization) in the text (Mat&Met or Results) rather than in the Table itself.
OFES definition should be provided as a footnote to the Table.
The percentage of participants and organizations should be provided in brackets rather than after the fraction sign ("/"). "(100%)" might be added for the two columns in the "Total" line.
Author Response
Hereby our point by point reaction on round 2 of reviewer 1
Despite the Authors' declaration that a list of questions used in the questionnaire was now provided, I couldn't see it in the new draft of the paper. I suggest providing either a summary description of the questions in the Mat&Met section, or a more detailed list as Supplementary Material. We have send a list of questions in the reaction round 1 but we will add the whole list of questions in the Supplementary materials as suggested by the reviewer
Table 1:
I suggest reporting the new details provided on farmers (i.e., herd size) and Veterinarians (i.e., dairy cattle specialization) in the text (Mat&Met or Results) rather than in the Table itself. Changed as suggested by reviewer
OFES definition should be provided as a footnote to the Table. Definition provided as footnote
The percentage of participants and organizations should be provided in brackets rather than after the fraction sign ("/"). "(100%)" might be added for the two columns in the "Total" line. Changed as suggested by reviewer
Kind regards,
Len Lipman